# Dynamic kernel matching for non-conforming data: A case study of T cell receptor datasets

**Jared Ostmeyer●\*, Lindsay Cowell, Scott Christley**

Department of Population and Data Sciences, University of Texas Southwestern Medical Center, Dallas, Texas, United States of America

\* Jared.Ostmeyer@UTSouthwestern.edu

**Data Availability Statement:** All data, links to data, and detailed instructions on how to access the data are provided at https://github.com/jostmey/dkm.

**Funding:** Yes: LC and SC, R01AI097403, U.S. Department of Health & Human Services | NIH |

## Abstract

Most statistical classifiers are designed to find patterns in data where numbers fit into rows and columns, like in a spreadsheet, but many kinds of data do not conform to this structure. To uncover patterns in non-conforming data, we describe an approach for modifying established statistical classifiers to handle non-conforming data, which we call *dynamic kernel matching* (DKM). As examples of non-conforming data, we consider (i) a dataset of T-cell receptor (TCR) sequences labelled by disease antigen and (ii) a dataset of sequenced TCR repertoires labelled by patient cytomegalovirus (CMV) serostatus, anticipating that both datasets contain signatures for diagnosing disease. We successfully fit statistical classifiers augmented with DKM to both datasets and report the performance on holdout data using standard metrics and metrics allowing for indeterminant diagnoses. Finally, we identify the patterns used by our statistical classifiers to generate predictions and show that these patterns agree with observations from experimental studies.

## Introduction

Statistical classifiers are mathematical models that use example data to find patterns in features that predict a label. The features and corresponding label are referred to as a sample. Standard statistical classifiers implicitly assume (i) each sample has the same number of features and (ii) each feature has a fixed input occupying the same position across all samples (Fig 1a). For example, the first feature might always represent a patient's age and the second feature might always represent their height. Under these assumptions, each input of the statistical classifier always gets features conveying the same kinds of information. We call features non-conforming when these assumptions do not hold, requiring specialized approaches to handle a lack of correspondence between the non-conforming features and the inputs of the statistical classifier (Fig 1b).

Sequences are one example of a datatype with non-conforming features. The essential property of a sequence is that both the content and the order of the symbols in the sequence convey information. Sequence data is non-conforming because some sequences are longer than others, resulting in irregular numbers of features. Even when sequences are the same length, a pattern of symbols shared between these sequences can appear at different positions, preventing

National Institute of Allergy and Infectious Diseases (NIAID), https://www.nih.gov/ JO, LC, and SC, 825821, EC | Horizon 2020 Framework Programme (EU Framework Programme for Research and Innovation H2020), https://ec.europa.eu/programmes/horizon2020/.

**Competing interests:** The authors have declared that no competing interests exist.

the same feature from appearing at the same position across all samples. To handle sequences, statistical classifiers need to be able to find patterns in either the content or the order of the symbols that predict the label. Shimodaria and colleagues developed an approach that does both [1]. Their statistical classifier uses weights to both tune how each feature contributes to the prediction and place each feature into context based on its order in the sequence. Their approach, originally developed for support vector machines, has been applied to deep neural networks by Iwana and Uchida [2, 3].

In this study, we adapt Iwana and Uchida's approach to handle a broader suite of non-conforming features, beginning with (mathematical) sets. The essential property of a set is that the content but not the order of the symbols conveys information, and therefore the order of the symbols has no meaning. Sets are examples of non-conforming data because there is no order indicating the position of each feature. By adapting their statistical classifier to also handle sets, our approach uses weights to both tune how each feature contributes to the prediction and place each feature into context based on the presence (but not the order) of other features. We go a step further, combining approaches for sequences and sets into a composite model for sets made of sequences, which uses the weights to tune how each feature contributes to the

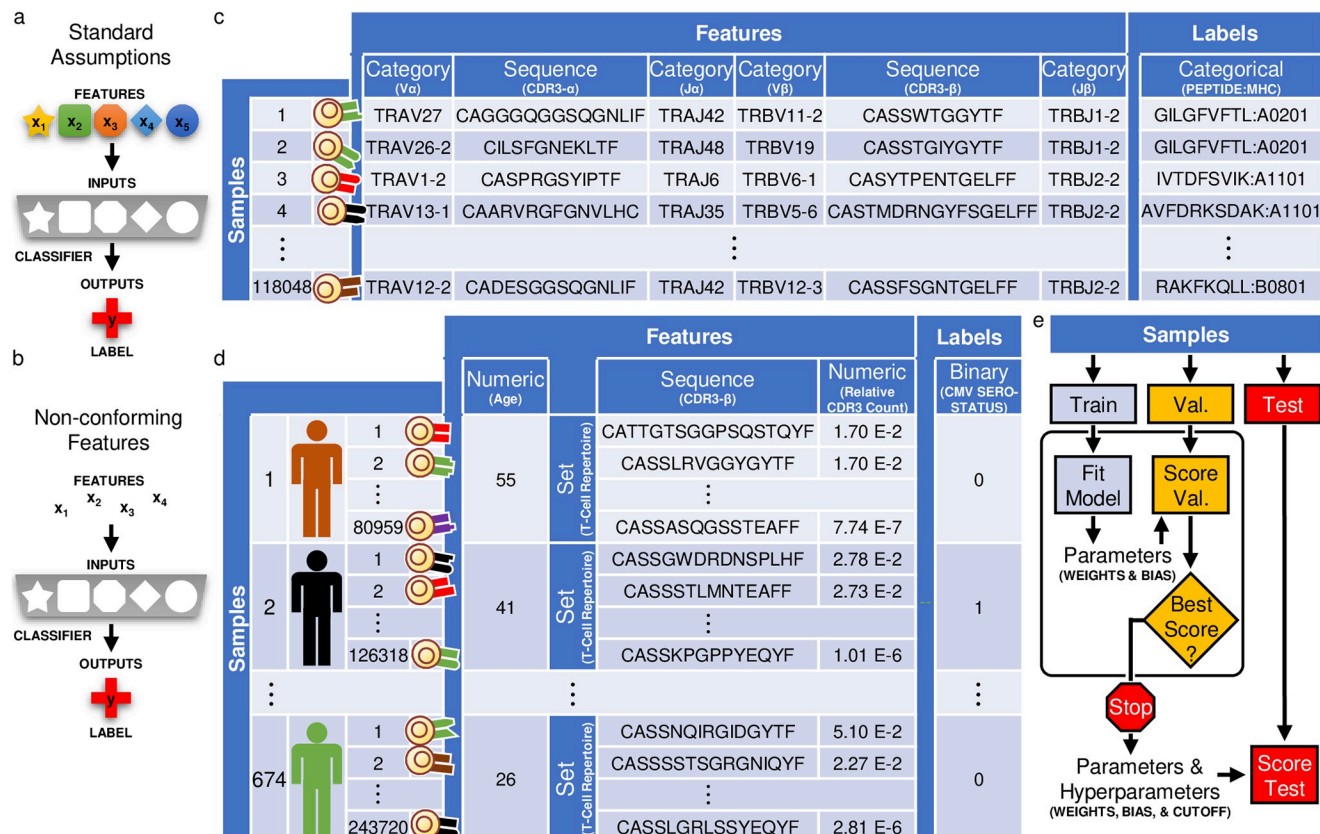

**Fig 1. TCRs are examples of non-conforming data.** (a) Most statistical classifiers assume a fixed number of features (e.g. five) and that each feature represents the same kind of information across samples (e.g. shape). (b) These assumptions do not hold for non-conforming data. (c) A dataset of TCRs labelled by interaction with disease antigens. The dataset contains the amino acid symbols from regions of the TCR (CDR3) represented as sequences, which are examples of non-conforming data. (d) A dataset of TCR repertoires labelled by CMV serostatus. The dataset contains sequenced TCR repertoires represented as sets made of sequences, which is a different kind of non-conforming data than the previous dataset. (e) Samples are split into a training, validation, and test cohort (for panel c, identical sequences are first collapsed to ensure the same TCR does not appear in multiple cohorts). The training cohort is used to select the weights and biases of each model, the validation cohort is used for model selection, and the test cohort is used for reporting results.

prediction and place each feature into context based on (i) its order in that sequence and (ii) the presence (but not the order) of other sequences in each set. We call our approach *dynamic kernel matching* (DKM), providing a unified approach for classifying sequences, sets, sets of sequences, and potentially other non-conforming data as well. DKM also represents an extension of our previous approach for classifying sets, which we used to discover various disease signatures in patients' TCR sequences, by alleviating constraints that forced us to use only a single fixed-length subsequence from the top scoring TCR sequence to predict each label [4–7].

As examples of non-conforming data, we consider datasets of T cell receptors (TCRs), anticipating these datasets to contain signatures for diagnosing disease. Other studies have avoided the problem of non-conforming features in TCR datasets by using handcrafted features, summary statistics, and preprocessing steps to reduce non-conforming data to conforming data, thereby producing conforming features that can be classified using standard methods [8–18]. While these strategies have been successful, there is no systematic approach that can guarantee essential information is not lost by reducing non-conforming data to conforming features. For this reason, approaches have been developed that directly classify non-conforming features rather than rely on a reduction step [1, 19–21]. Most of these approaches assume the non-conforming features follow a temporally directed relationship where the past contextualizes the present, but the present cannot contextualize the past. These assumptions may be inappropriate for datasets of TCRs, which do not necessarily contain notions of past and present. Because DKM does not impose notions of past and present, DKM may be ideal for datasets of TCRs. In this study, we showcase DKM on datasets of TCRs containing (i) sequences or (ii) sets of sequences, depending on what is labelled.

To demonstrate DKM's performance on classifying sequences, we fit (i) a multinomial regression model augmented with DKM to sequenced TCRs labelled by disease antigen (Fig 1c) [22]. To demonstrate DKM's performance on classifying sets of sequences, we fit (ii) a logistic regression model augmented with DKM to sequenced TCR repertoires labelled by cytomegalovirus (CMV) serostatus (Fig 1d) [5]. We then test the performance of our statistical classifier on samples from blindfolded cohorts (Fig 1e). Finally, we analyze the statistical classifiers in the context of existing experimental data and discover these models may have identified biologically relevant patterns in the data. Our results collectively demonstrate DKM can be used to potentially uncover meaningful patterns in non-conforming features that predict the labels.

## Dynamic kernel matching

### Representing non-conforming features

As with any classification problem, the first step is to decide upon a way to represent the data, which in this study includes non-conforming features. We assume that each sample of non-conforming features consist of multiple symbols organized into some sort of structure, such as a sequence or a set.

We represent each symbol using a vector of $N$ numbers. For example, if we have a sequence of symbols, we replace each symbol by its vector representation, resulting in a sequence of vectors. Each time the same symbol appears, we use the same vector of numbers to represent that symbol, providing a consistent representation for each symbol across the dataset. Ideally, the numbers in each vector should describe properties of that symbol. In this study, the symbols represent amino acids and the numbers in each vector describe physicochemical properties of the amino acids, providing information necessary to determine, for example, that symbols R and K represent amino acids with positively charged sidechains interchangeable with respect

to this common property (specifically, we use the five Atchley numbers to represent each amino acid [23]). We use $i$ to index the symbols, $\vec{x}_i$ to represent a specific symbol indexed by $i$, and $[x_{i,1}, x_{i,2}, \ldots, x_{i,N}]$ to represent the $N$ numbers in $\vec{x}_i$ (S1a Fig).

## Representing weights

Suppose the statistical classifier is a linear regression model and therefore has weights for multiplying with features (DKM can also be used with a support-vector machine or deep neural network).

Because we are dealing with non-conforming features, the number of features is irregular and need not correspond to the number of weights. Therefore, we can pick the number of weights in our model irrespective of the number of features. We pick the number of weights to be evenly divisible by $N$ *(as defined in the previous section)*, allowing us to group the weights into vectors of $N$ weights. We use $j$ to index these vectors, $\vec{\theta}_j$ to refer to a specific vector indexed by $j$, and $[w_{j,1}, w_{j,2}, \ldots, w_{j,N}]$ to represent the $N$ weights contained by a vector $\vec{\theta}_j$ (S1b Fig).

If we can determine a specific vector $\vec{x}_i$ to match to a specific vector $\vec{\theta}_j$ then we can assign the features in $\vec{x}_i$ to the weights in $\vec{\theta}_j$ because $\vec{x}_i$ and $\vec{\theta}_j$ are the same size $N$.

## Similarity score

To determine if a specific vector $\vec{x}_i$ should be matched to a specific vector $\vec{\theta}_j$, we need a *similarity score* to measure the similarity between these vectors. We define the similarity score between $\vec{x}_i$ and $\vec{\theta}_j$ as the sum of the features in $\vec{x}_i$ multiplied by the weights in $\vec{\theta}_j$.

$$S(\vec{x}_i, \vec{\theta}_j) = \sum_{k=1}^{N} x_{i,k} \cdot w_{j,k} \tag{1}$$

Using Eq (1) to calculate each similarity score, we will match multiple vectors to maximize the overall similarity scores. Because each similarity score depends on the weights, the weights influence which $\vec{x}_i$ are matched to $\vec{\theta}_j$. As we will see, the algorithm for maximizing the overall similarity scores is different for sequences than sets, but after that the subsequent steps are the same regardless of which maximization algorithm is required for the datatype.

## Maximizing the sum of similarity scores for sequences

Suppose that the symbols in each sample represent a sequence. In this case, we can use a *sequence alignment algorithm* to match each $\vec{x}_i$ to each $\vec{\theta}_j$. Sequence alignment algorithms pad sequences to place similar symbols from two sequences into matching positions. Similarity for each possible match is determined using a similarity score, like defined in Eq (1). The sum of the similarity scores across every position is called the *alignment score*, and the objective of a sequence alignment algorithm is to find how to match symbols in the sequences to maximize this alignment score. Many sequence alignment algorithms have been developed, such as the Needleman–Wunsch algorithm used in this study, that guarantee finding an alignment with the maximum possible alignment score [24].

## Maximizing the sum of similarity scores for sets

Suppose that the symbols in each sample represent a set. In this case, we can solve the *assignment problem* to match each $\vec{x}_i$ to each $\vec{\theta}_j$. We can think of solving the assignment problem as

running an alignment algorithm, like before, except now the order of the symbols does not matter. Similarity for each possible match is determined using a similarity score, like defined in Eq (1). We refer to the sum of the similarity scores between matched symbols as an alignment score, and the objective of the assignment problem is to maximize this alignment score. Algorithms exist for solving assignment problems, like the Hungarian method [25].

## Maximizing the sum of scores for sets made of sequences

Suppose that the symbols in each sample represent a set made of sequences. In this case, we can combine the two previous algorithms to match each $\vec{x}_i$ to each $\vec{\theta}_j$. First, we match each $\vec{x}_i$ to each $\vec{\theta}_j$ using a sequence alignment algorithm. The similarity score is defined using Eq (1). The sum of the similarity scores produced by aligning sequences is treated as a *sub*-alignment score. The sub-alignment scores serve as the similarity scores in the assignment problem, allowing us to match sequences. We refer to the sum of the similarity scores in the assignment problem as the alignment score, and the objective of the assignment problem is the same as maximizing this alignment score.

## Generating predictions

Many statistical classifiers generate a prediction by calculating the sum of the features multiplied by the weights. We will see how we can reuse the alignment score from any of the previous maximization algorithms to arrive at a similar calculation.

Because the alignment score is the sum of similarity scores over the matched symbols, we need to introduce sub-indices to indicate the matches. For example, if $\vec{x}_1$ is unmatched but $\vec{x}_2$ is matched to $\vec{\theta}_3$ then we use sub-indices $i_1 = 2$ and $j_1 = 3$ to represent the 1$^{\text{st}}$ match. If $\vec{x}_3$ is matched to $\vec{\theta}_4$ then we use sub-indices $i_2 = 3$ and $j_2 = 4$ to represent the 2$^{\text{nd}}$ match. Using sub-indices and letting $L$ represent the number of matches, we can represent the matches produced by an alignment algorithm.

$$
\begin{matrix}
\vec{x}_{i_1} & \vec{x}_{i_2} & \cdots & \vec{x}_{i_L} \\
| & | & & | \\
\vec{\theta}_{j_1} & \vec{\theta}_{j_2} & \cdots & \vec{\theta}_{j_L}
\end{matrix}
$$

We can now write the alignment score as the sum of the similarity scores using the sub-indices.

$$
A = \sum_{l=1}^{L} S(\vec{x}_{i_l}, \vec{\theta}_{j_l}) = \sum_{l=1}^{L} \sum_{k=1}^{N} x_{i_l,k} \cdot w_{j_l,k} = \sum_{l=1,k=1}^{L,N} x_{i_l,k} \cdot w_{j_l,k} \tag{2}
$$

$A$ denotes the alignment score. On the left side of Eq (2), the alignment score is written as the sum of similarity scores $S(\vec{x}_{i_l}, \vec{\theta}_{j_l})$ over the matches. Each similarity score is written as a sum of the features multiplied by the weights using Eq (1), resulting in a double summation. The double summation is flattened into a single summation on the right side of Eq (2), resulting in a summation of features multiplied by weights. Thus, the alignment score is a sum of features multiplied by weights, representing the same calculation required by a linear regression model to arrive at a prediction.

It is tempting to use the unnormalized value of the alignment score $A$ to classify each sample because the right side of Eq (2) is a sum of features multiplied by weights. However, the magnitude of $A$ can vary depending on the number of matches, which can be undesirable for

some classification problems. If we assume the variances of the similarity scores at each of the $L$ positions are roughly the same, then the variance of $A$ proportionally varies by $\geq L$. Therefore, we normalize $A$ by dividing by the lower bound of the standard deviation, $\sqrt{L}$, using this value to classify $X$. For example, if we use logistic regression for the binary classification of a sequence, the logit would be defined as $\text{logit} = A/\sqrt{L}$ and the predicted probability would be $P = 1/(1 + e^{-\text{logit}})$. Because we normalize away information about $L$, we include $L$ as a feature in the statistical classifier. We also include a bias term $b$ in each logit, which is a free parameter that sets the baseline probability when all the features have a value of zero.

Given a method to generate a prediction, we can define the objective function for the problem as would be done for any other statistical classifier. Gradient optimization techniques can be used to find values for the weights and bias term that maximize the probability of achieving the correct predictions for each sample in a training cohort [26].

See "S1 File, *Dynamic Kernel Matching*" for parameter fitting and other details.

## Confidence cutoffs

Biomarker studies have used rule-in and rule-out cutoffs to diagnose patients with and without a condition. Patients not captured by these cutoffs are given an indeterminate diagnosis indicating the need for additional observation or exams. By providing an indeterminate diagnosis on uncertain cases, only the patients that can be diagnosed with a high degree of confidence receive a diagnosis, resulting in a higher classification accuracy [27, 28]. When conducting a blindfolded study, the labels for uncaptured samples must remain blindfolded. The classification accuracy is calculated only from the unblindfolded samples captured by the rule-in and rule-out cutoffs. When reporting the classification accuracy, the number of samples captured by the cutoffs must be reported too. If the number of captured samples is not reported, all samples are assumed to be used in the calculation of the classification accuracy.

Rather than use rule-in and rule-out cutoffs, we introduce an entropy-based cutoff, which is not limited to just binary classification problems but can be applied to multinomial and regression problems as well. With this approach, entropy is used to measure the confidence of each prediction, as defined below.

$$H_j = -\sum_{i=1}^{M} p_j^{(i)} \cdot \ln p_j^{(i)}$$

$H_j$ represents the entropy associated with the prediction for the $j^{\text{th}}$ sample, $M$ represents the number of label categories, $i$ indicates a specific label category, and $p_j^{(i)}$ represents the probability assigned to each category by the statistical classifier. We define $H_{\text{cutoff}}$ as the cutoff for capturing samples. If $H_j \leq H_{\text{cutoff}}$ the sample is classified because the confidence is high. Otherwise, the sample is not captured by the cutoff because the confidence is low. In this study, we start with a value for $H_{\text{cutoff}}$ large enough to ensure all the samples are initially captured and decrease $H_{\text{cutoff}}$ in increments of 0.01 until we find that the accuracy over captured samples is $\geq 95\%$ on a validation cohort. We then apply the cutoff to capture samples on a blindfolded test cohort, unblind ourselves to the captured samples, and compute the accuracy.

## Datasets

Datasets of TCR sequences provide examples of non-conforming features for diagnosing disease. Classifying TCRs requires handling (i) sequences or (ii) sets of sequences, depending on what is labelled.

### Antigen classification dataset

Individual TCRs can be labelled by the disease-relevant molecules it can bind, called antigens, consisting of a peptide presented on a major histocompatibility complex (pMHC). Because each TCR gene is generated by random insertions and deletions in what is called complimentary determining regions 3 (CDR3s) located between stochastically paired V- and J-gene segments, each TCR cannot be adequately characterized by its V- and J-genes without the CDR3 sequences as well. To solve what we call the *antigen classification problem*, we need a statistical classifier that can handle the CDR3 sequences of each TCR. 10x Genomics has published a dataset of sequenced TCRs barcoded by a panel of pMHCs (arranged on a dextramer) (Fig 1c) [22]. The goal is to predict the pMHC from the sequenced TCR, requiring an approach to handle the sequences of amino acid symbols in the CDR3s. Of the 44 pMHCs in the dataset, only the six pMHCs interacting with at least 500 unique receptors are used for the antigen classification problem, which are GILGFVFTL:A0201 (3 982 samples), KLGGALQAK:A0301 (27 561 samples), RLRAEAQVK:A0301 (612 samples), IVTDFSVIK:A1101 (2 563 samples), AVFDRKSDAK:A1101 (3 505 samples), and RAKFKQLL:B0801 (5 598 samples). Because this dataset is imbalanced, samples have been weighted to represent each of the six pMHCs equally. The dataset is then split into a training, validation, and test cohort using a 60/20/20 split. See S1 File for details.

### Repertoire classification dataset

In circumstances where individual TCRs are not labelled by disease antigen, we can still label TCR sequences by the patient's disease status. Patterns exist that predict this label because a patient's set of TCR sequences, referred to as the TCR repertoire, constantly adjust to the presence of disease antigens, and therefore contain traces of past and ongoing immune responses to the underlying diseases. To solve what we call the *repertoire classification problem*, where the patients are labelled but the individual TCRs are not, we need a statistical classifier that can handle the set of CDR3 sequences in the TCR repertoire. Adaptive Biotechnologies has published a dataset of TCR repertoires sequenced from peripheral blood and labelled by CMV serostatus (Fig 1d) [5]. The goal is to predict each patient's CMV serostatus from their TCR repertoire, requiring an approach to handle the set of CDR3 sequences from each patient. Adaptive Biotechnologies provides the dataset as Cohort I and II, the latter being designated as the test cohort. For this study, Cohort I (554 patients) is randomly shuffled and split into a training (434 patients) and validation cohort (120 patients) and Cohort II (120 patients) is used as the test cohort. Of the 674 total patients, 309 have a CMV+ serostatus and 365 have a CMV- serostatus. Because this dataset is imbalanced, samples have been weighted to represent CMV+ and CMV- cases equally. See S1 File for details.

## Results

### Antigen classification problem

To demonstrate that a statistical classifier augmented with DKM can classify sequences, we modify a multinomial regression model (S2 Fig) and fit it to the antigen classification dataset (Fig 1c) [22]. The dataset has been balanced to represent each category equally. At every gradient optimization step, the fit (as measured by KL-divergence) to samples from the training cohort steadily improves from 2.71 to 0.684 bits, demonstrating that even a multinomial regression model augmented with DKM can fit complex data (to put these numbers in perspective, six balanced outcomes like the categories in this dataset have an entropy of $\log_2 6 \approx 2.58$ bits). Measuring the fit to samples from the validation cohort at each gradient

optimization step reveals a steady improvement from 2.60 to 1.18 bits, demonstrating the statistical classifier generalizes to holdout samples not used to fit the weight and bias values (Fig 2a). At the end of the study, we unblindfold ourselves to samples from the test cohort and measure the fit as 1.31 bits, consistent with results from the validation cohort (Fig 2a).

Next, we determine the balanced classification accuracy on samples from the test cohort (without confidence cutoffs). We calculate the balanced classification accuracy as the average number of times the most probable prediction matches the true label, averaged over all samples. On the test cohort, the accuracy is 70.5%. Because there are six possible outcomes, the baseline accuracy achievable by chance compares to guessing the outcome of a six-sided die roll. A confusion matrix between the predictions and labels for the test cohort reveals strong agreement along the diagonal (Fig 2b).

We hypothesize the statistical classifier generates accurate predictions using specific non-conforming features describing molecular interactions between the TCR and pMHC. To attempt to answer this question, we search existing 3D X-ray crystallographic structures of TCR bound to any of the six pMHCs in the dataset, which would allow us to analyze the statistical classifier in the context of the observed molecular interactions between the TCR and pMHC. We find four unique TCRs bound to GILGFVFTL:A0201, a pMHC in our dataset [29–31]. The statistical classifier correctly predicts the four TCRs interact with this pMHC, but only two of the TCRs are captured by ≥95% confidence cutoffs (S3a Fig.). We restrict our analysis to these two receptors, anticipating these cases to exhibit the clearest results. To ascertain the importance of each non-conforming feature, we conduct an in-silico alanine scan by systematically replacing each amino acid symbol in the CDR3s with alanine (symbol A), computing the change in logit, and then putting back the original amino acid symbol into the CDR3. The greatest logit changes tend to correspond to positions in the CDR3 in contact with pMHC (≥5Å), with the greatest logit change always corresponding to a contact position, as indicated by asterisks (Fig 2c, S3b and S3c Fig). These observations agree with the hypothesis that the statistical classifier identifies and utilizes non-conforming features describing the molecular interactions between the TCR and pMHC.

To compare the classification accuracy of DKM to the accuracy achievable with other approaches, we run other statistical classifiers on the antigen classification problem. For example, a method named TCRDist provides a distance metric for TCRs, making it possible to generate predictions for each TCR using a nearest neighbor classifier [12]. This approach achieves a classification accuracy of 67.9% on the test cohort. The distances between TCR sequences are calculated in part with a sequence alignment algorithm. While TCRDist and DKM both use sequence alignment algorithms, the similarity scores for TCRDist are taken from the BLOSUM62 matrix whereas the similarity scores for DKM are determined by the weights of the model. Another strategy for classifying TCRs is to use handcrafted features. De Neuter and colleagues developed handcrafted features for TCR sequences that includes amino acid frequency, positional encodings of each amino acid, and the physicochemical properties from each amino acid averaged together [13]. Using De Neuter's handcrafted features and a random forest model we achieve a classification accuracy of 67.4% on the test cohort. We also fit recurrent and convolutional neural networks that can classify raw sequences without the need for handcrafted features, but these deep learning approaches achieved some of the lowest classification accuracies. See S4 Fig for details (S4 Fig). To summarize, none of the approaches in our comparison had a better classification accuracy than DKM.

## Repertoire classification problem

To demonstrate that a statistical classifier augmented with DKM can classify sets of made sequences, we modify a logistic regression model (S5 Fig) and fit it to the repertoire

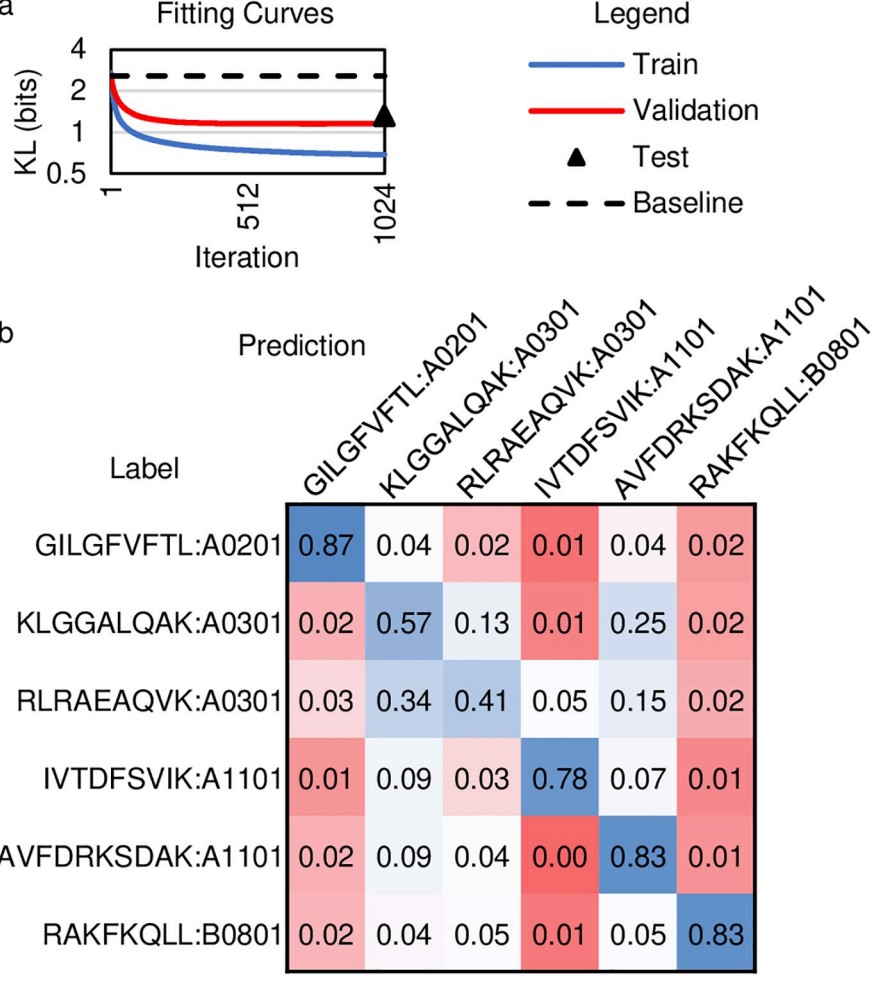

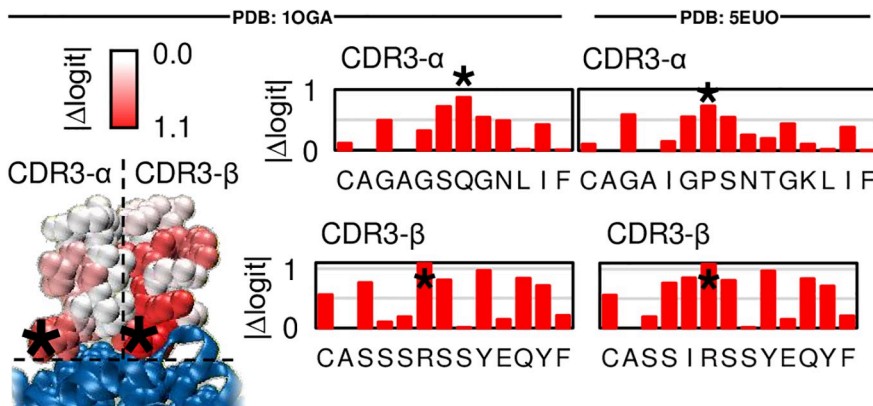

**Fig 2. Results on the antigen classification problem.** (a) The fit plotted for each weight update across the training (solid blue) and validation cohorts (solid red) steadily improves with each gradient optimization step. The fit to the test cohort after unblinding the samples (triangle) is significantly better than the baseline performance achievable by random chance (dashed black). (b) A confusion matrix of samples from the test cohort reveals the fraction of predictions that agree with the labels for each category (c) A 3D X-ray crystallographic structure of a TCR bound to GILGFVFTL:A0201. An alanine scan of the TCR CDR3 sequences reveals the largest |Δlogit| always corresponds to a contact position (asterisks).

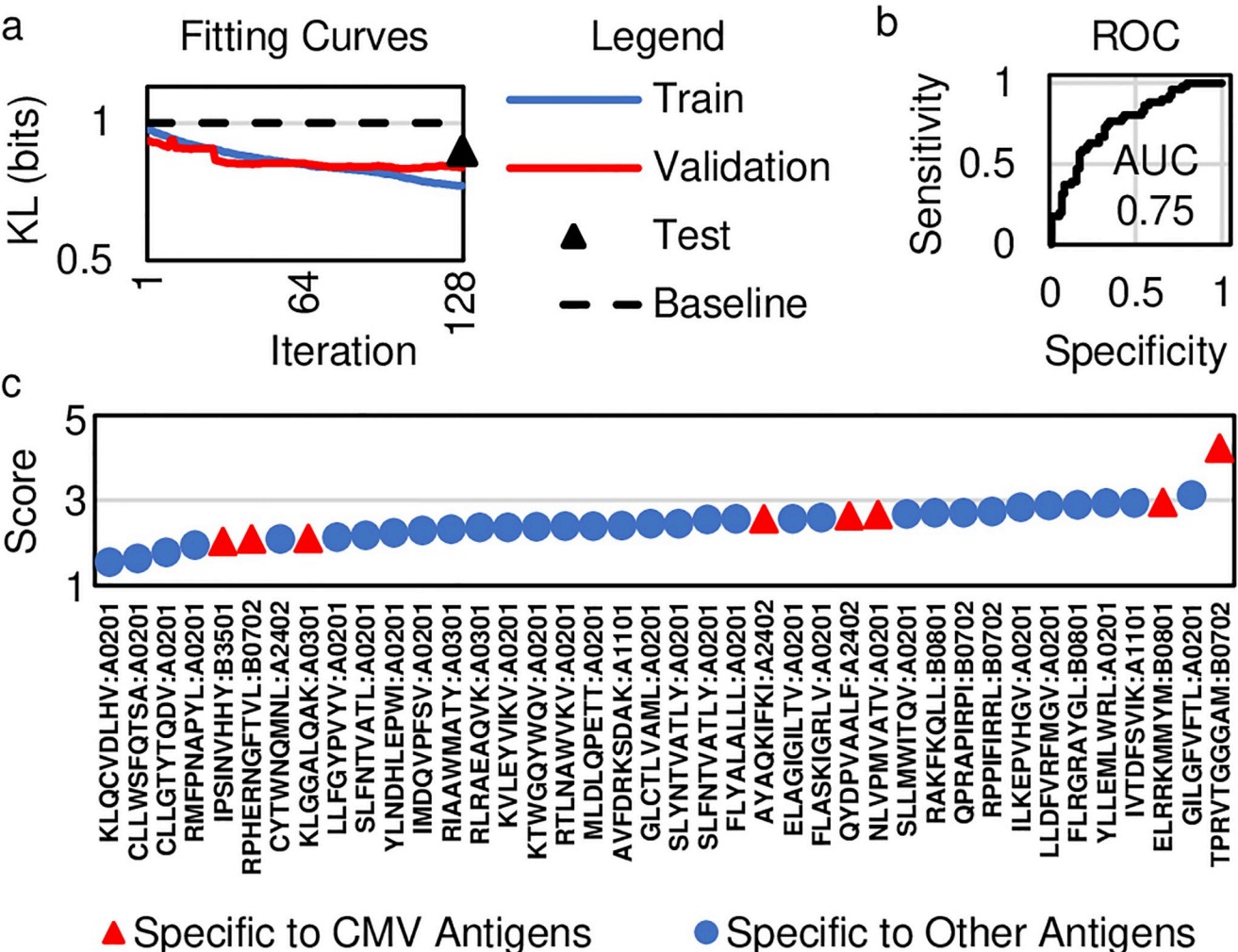

**Fig 3. Results on the repertoire Classification problem.** (a) The fit plotted for each weight update across the training (solid blue) and validation cohorts (solid red) steadily improves with each gradient optimization step. The fit to the test cohort after unblinding the samples (triangle) is better than the baseline performance achievable by random chance (dashed black). (b) A ROC curve reveals the sensitivity and specificity for various diagnostic thresholds of the model. (c) CDR3 sequences from receptors specific to various pMHCs are scored using the kernel of the model fitted in panel a. The receptors specific to a CMV peptide have the highest score, suggesting that the model has learned to identify receptors specific to this CMV peptide.

classification dataset (Fig 1d) [5]. The dataset has been balanced to represent each category equally. At every gradient optimization step, the fit (as measured by KL-divergence) to samples from the training cohort steadily improves from 0.966 to 0.728 bits, demonstrating a logistic regression model augmented with DKM can fit even this highly complex dataset (*we refit the model 128 times in an attempt to find the global optimum, using the best fit as measured on samples from the training cohort,* S5j Fig). Measuring the fit to samples from the validation cohort at each gradient optimization step reveals a steady improvement from 0.914 to 0.798 bits, demonstrating the statistical classifier generalizes to holdout samples not used to fit the weight and bias values (Fig 3a). At the end of the study, we unblindfold ourselves to samples from the test cohort and measure the fit as 0.869 bits, consistent with results from the validation cohort (Fig 3a).

Next, we determine the balanced classification accuracy on samples from the test cohort (without confidence cutoffs). We calculate this classification accuracy as the average number

of times the most probable prediction matches the true label, balanced over the two categories (i.e. CMV+ and CMV-). On the test cohort, the accuracy is 67.6%. A plot of the true versus false positive rates for various diagnostic thresholds of the model, known as a receiver operating characteristic (ROC) curve, has an area under the curve (AUC) of 0.75 (Fig 3b).

We hypothesize the statistical classifier generates accurate predictions using specific non-conforming features describing TCRs that bind to CMV peptide. To answer this question, we use the weights and bias values to score β-chain CDR3 sequences in the dataset published by 10x Genomics, allowing us to score and compare TCR CDR3 sequences interacting with CMV peptide to those interacting with non-CMV peptides (*essentially, the kernel of the statistical classifier fitted without individually labelled TCRs is being used to score TCRs interacting with known pMHC*). For features from the repertoire classification problem not found with individual CDR3 sequences, like patient age, we use a value of zero for the incomplete data. Because these features are assigned a value of 0, we can only evaluate the scores relative to other scores in the 10x Genomics dataset. For each pMHC in the 10x Genomics dataset, we average the scores for CDR3 sequences from TCRs that interact with the same pMHC. TPRVTGGGAM: B0702 has the largest score, which is a CMV peptide, congruent with our hypothesis that the statistical classifier identifies and utilizes non-conforming features describing TCRs that interact with CMV peptide (Fig 3c). In fact, two of the top three scoring pMHCs contain CMV peptide. Other CMV peptides do not have comparably high scores, indicating the statistical classifier, fitted without using labels on individual TCRs, cannot identify receptors interacting with every possible CMV peptide.

Comparing the classification accuracy of DKM to the accuracy achievable by other approaches on the repertoire classification problem, we find our result is significantly worse than a previously reported result by Emerson and colleagues [5]. However, their approach is specific for the repertoire classification problem and cannot be applied to the antigen classification problem, and the source code to reproduce their results has not been made available for review. We also applied our previous approach for solving the repertoire classification problem [4–7] and achieved a classification accuracy of 68.3% on the test cohort, similar to what we achieve with DKM. See S6 Fig for details (S6 Fig).

## Applying confidence cutoffs

Even with a low classification accuracy, a statistical classifier can prove useful if it provides accurate positive and negative diagnoses for a subset of patients. With this goal in mind, we use confidence cutoffs to capture a subset of samples that can be classified with ≥95% accuracy. We find that cutoffs of $H_{\text{cutoff}}^{\text{val}} = 0.98$ and $H_{\text{cutoff}}^{\text{val}} = 0.527$ for the antigen and repertoire classification problems, respectively, achieve classification accuracies of ≥95% over samples captured from the validation cohort. Next, we run samples from the test cohort through the statistical classifiers, applying the confidence cutoffs $H_{\text{cutoff}}^{\text{val}}$ to capture samples from the test cohort. The labels are unblindfolded only for captured samples, and the classification accuracy is computed. On the antigen classification problem, the classification accuracy is 97.1% capturing 44.5% of test cohort samples, and on the repertoire classification problem, the classification accuracy is 96.0% capturing 18.0% of test cohort samples (Fig 4, S7 Fig).

We wondered if the confidence cutoffs capture only samples from a restricted subset of categories or if samples are captured evenly across all categories. On the antigen classification problem, captured samples are not balanced across categories (Fig 4a). Nearly every sample for categories KLGGALQAK:A0301 and RLRAEAQVK:A0301 are left uncaptured. These two pMHCs represent the most challenging cases to distinguish given that the peptides are the same length sharing biochemically similar amino acid residues presented on the same MHC,

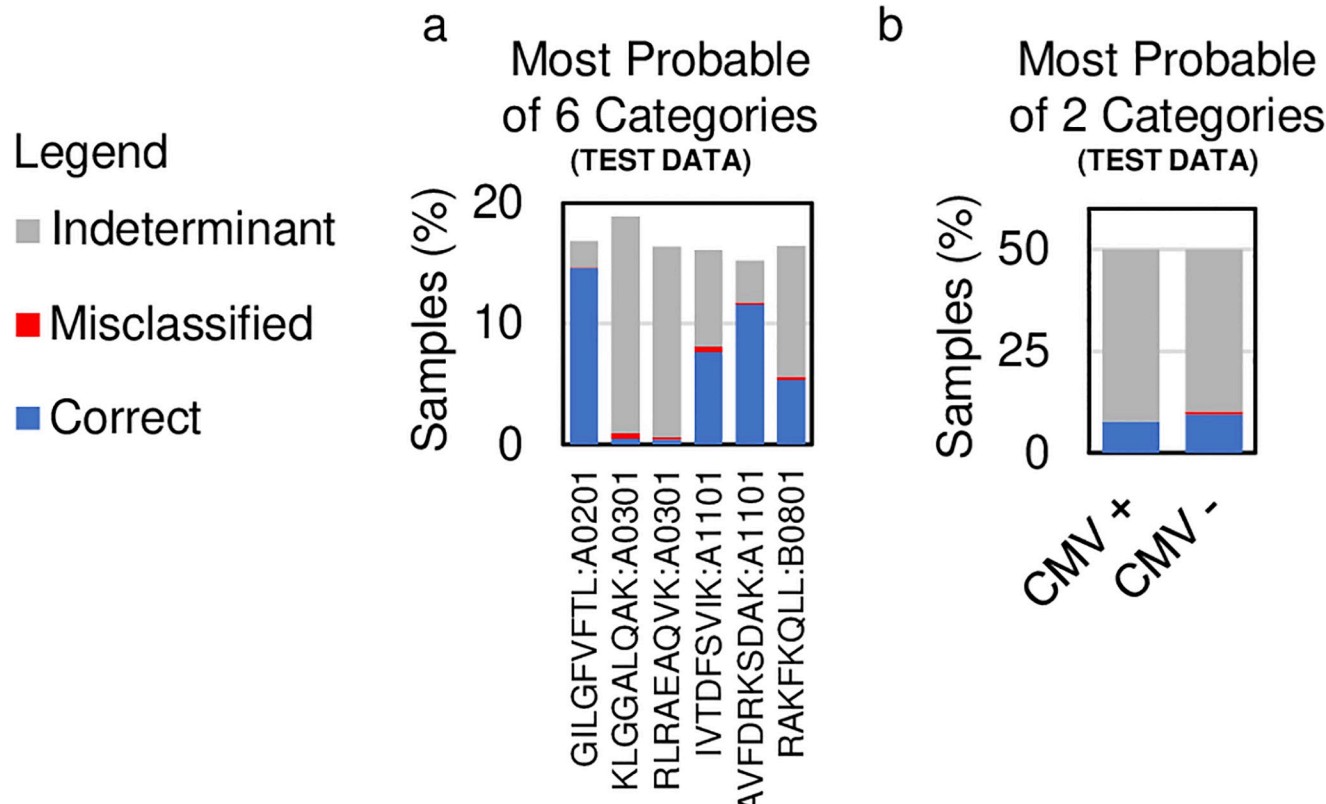

**Fig 4. Results with confidence cutoffs.** The bar charts show the fraction of samples that are correctly classified (blue), incorrectly classified (red), and indeterminate (grey) (a) Using a 95% confidence cutoff determined on the validation cohort, the classification accuracy on the test cohort of the antigen classification problem is 97.06% capturing 44.5% of samples. The samples are not captured evenly across the six categories. (b) Using a 95% confidence cutoff determined using the validation cohort, the classification accuracy on the test cohort of the repertoire classification problem is 96.0% capturing 18.0% of samples. The samples are captured almost evenly across the two categories.

explaining why the confidence in the predictions is not high enough to capture samples from these categories. On the repertoire classification dataset, the captured samples are approximately balanced across the two categories representing CMV+ and CMV- samples (Fig 4b). Thus, whether captured samples evenly include all categories appears to depend on whether one category is more challenging to distinguish than others.

## Discussion

We demonstrate DKM can handle sequences, fitting a multinomial regression model augmented with DKM to sequenced TCRs labelled by interaction with disease antigen. The fitted model generalizes to samples in the test cohort, predicting the antigen with 70.5% accuracy, which is well above the 16.6% classification accuracy achievable by random chance and better than the alternative approaches we tried (S4 Fig). To generate accurate predictions, we hypothesize the model must be using non-conforming features that describe molecular interactions between the TCR and pMHC. Using the sequences of TCRs in 3D X-ray crystallographic structures, we identify the amino acid residue in each CDR3 sequence most important to the correct prediction. We observe that this residue is always in contact with pMHC ($\geq 5$Å). These results are consistent with the hypothesis that DKM uses non-conforming features describing molecular interactions between the TCR and pMHC to generate accurate predictions.

Next, we demonstrate DKM can handle sets of sequences, fitting a logistic regression model augmented with DKM to sequenced TCR repertoires labelled by cytomegalovirus (CMV) serostatus. We essentially reuse the DKM component for the antigen classification problem, having demonstrated that it handles the CDR3 sequences, and combine it with an approach for handling sets, to create an approach to solve the repertoire classification problem. On a test cohort, the fitted model achieves a classification accuracy of only 67.6%. To address this low accuracy, we use confidence cutoffs to capture samples that can be accurately diagnosed, excluding predictions where the uncertainty is too high. Using confidence cutoffs, approximately 8% of patients receive a CMV+ diagnosis, 10% receive a CMV- diagnosis, and 82% receive an indeterminate diagnosis. For the 18% of patients that are diagnosed, the classification accuracy is 96%. To generate accurate predictions, we hypothesize the model must be identifying TCRs specific to CMV antigen. To test this hypothesis, we score CDR3 sequences from TCRs interacting with known pMHCs using the weights of the fitted model. We observe that TCRs interacting TPRVTGGGAM, a CMV peptide, receive the highest score. These results are consistent with the hypothesis that the model uses TCRs specific to at least this CMV antigen to generate accurate predictions.

We find it remarkable that DKM can be used to build models to classify both sequences and sets of sequences, suggesting DKM can be reused on other datasets irrespective of the kind of data. We envision DKM providing a unified approach for handling many other kinds of non-conforming data and look forward to collaborating with other researchers to help build statistical classifiers for their problems.

## Materials and methods

Detailed instructions for downloading the data and building the datasets are included in the README of the source code accompanying this manuscript and available at http://github.com/jostmey/dkm. Briefly, data for the antigen classification problem can be downloaded directly from 10x Genomics webpage after creating a free account with 10x Genomics, and data for the repertoire classification problem can be downloaded from https://doi.org/10.21417/B7001Z after creating a free account with Adaptive Biotechnologies.

Downloaded sequencing data is used without imposing additional quality controls. The default threshold for the read counts provided in the 10x Genomics dataset are used (although the optimal value for this threshold is a matter of debate because the threshold effects the false and true positive rates as well as the number of TCRs assigned to each pMHC). Descriptions for shuffling and splitting the samples into training, validation, and test cohorts is available in the S1 File accompanying this manuscript and can be inferred from the source code.

The statistical classifiers in this study are implemented in TensorFlow v1.14 and run on a pair of RTX 2080 Ti GPUs or a cluster of either 8 P100 or V100 GPUs. Source code for the statistical classifiers accompanies this manuscript and is available at http://github.com/jostmey/dkm.

The 3D X-ray crystallographic structures analyzed in this study can be found at https://www.rcsb.org/ using the protein databank identifiers 1OGA, 5EUO, 5E6I, and 5ISZ. Renderings of the protein structures depicted in the figures are generated using Visual Molecular Dynamics (VMD).

## Supporting information

**S1 Fig. Representation of the variables for the non-conforming features and weights of a DKM model.** (a) Let us assume the non-conforming features consists of a sequence (or set) of $T$ symbols $x_1, x_2, \ldots x_i, .. x_T$. We replace each symbol with a vector of $N$ numbers

describing that symbol, resulting in a sequence (or set) of $T$ vectors $\vec{x}_1, \vec{x}_2, \ldots \vec{x}_i, \ldots \vec{x}_T$. (b) Let us assume the number of weights for our statistical classifier can form $R$ groups of $N$ weights. Each group of $N$ weights is used to form a vector, allowing us to write the weights as the vectors $\vec{\theta}_1, \vec{\theta}_2, \ldots \vec{\theta}_j, \ldots \vec{\theta}_R$. We can think of each vector as a symbol, forming the symbols $\theta_1, \theta_2, \ldots \theta_j, \ldots \theta_R$.
(TIF)

**S2 Fig. Schematic representation of the multinomial regression model augmented with DKM for the antigen classification problem.** (a) Features for each TCR are partitioned into six groups. (b) The representations for each feature group. (c) Dense modules assign a weight to each feature and compute a dot product. The DKM modules use a sequence alignment algorithm to match features with weights and compute a dot product. (d) Each dot product is normalized over training samples. The layout is repeated six times, one for each category of label. (e) The scaled dot products are added together into one dot product. (f) The resulting dot product is normalized over training samples. (g) The dot product is part of a multinomial logit and passed through a softmax function. (h) The softmax function produces probabilities representing the predictions of the statistical classifier. The six probabilities, one for each category, always sum to one.
(TIF)

**S3 Fig. Results for the 3D X-ray crystallographic structures found with a TCR bound to the antigens in the antigen classification problem.** (a) Four 3D X-ray crystallographic structures found with a TCR bound to GILGFVFTL:A0201. All four TCRs are correctly classified and two with $\geq 95\%$ confidence (see Applying Confidence Cutoffs). (b) An alanine scan of the 3D X-ray crystallographic structure 1OGA (four letter codes are the protein databank identifier at https://www.rcsb.org/) reveals the |Δlogit| tend to be greater (redder) for pMHC (green) contact positions ($\leq 5$Å) than non-contact positions. The largest |Δlogit| for each CDR3 is always a contact position. (c) Same as before for another 3D X-ray crystallographic structure 5EUO.
(TIF)

**S4 Fig. Comparison of statistical classifiers on the antigen classification problem.** Performance is measured using (i) KL-divergence, (ii) area under the curve (AUC) for true over false positive rates, and (iii) balanced classification accuracies. All models are fitted to the training cohort and results reported on the test cohort. The considered models are NN (nearest neighbors), RF (random forest), MR (multinomial regression), DNN (deep neural network), RNN (recurrent neural network), CNN (convolutional neural network), and DKM (dynamic kernel matching). DKM models have the best KL divergence fit, the highest AUC, and the highest balanced classification accuracies over the test cohort. The DKM model with the asterisk is the model reported in the main text. We performed a post-hoc hyperparameter optimization of our DKM model and found that we could achieve better results classifying each CDR3 as a sequence of 5-mers. However, we had already unblindfolded ourselves to the test cohort, so we did not report this result in the main text.
(TIF)

**S5 Fig. Schematic representation of the logistic regression model augmented with DKM for the repertoire classification problem.** (a) Features for each patient are partitioned into three groups. The last two feature groups, representing the CDR3-β sequence and relative CDR3 frequency, form a set. Each member of the set is passed through the model. (b) The representations for each feature group. (c) Dense modules assign a weight to each feature and

compute a dot product. The DKM modules use a sequence alignment algorithm to match features with weights and compute a dot product. (d) Each dot product is normalized over training samples. (e) The scaled dot products are added together into one dot product. (f) This DKM module takes the maximum value over the set, which is the computation required to solve the assignment problem for the special case where \Theta contains just one sequence (g) The maximum value is normalized over training samples. (h) The scaled value is passed through a sigmoid function. (i) The sigmoid function produces a probability representing the prediction of the statistical classifier. (j) Random values for each weight are refined by 128 steps of gradient optimization to produce a fitted model. Weights from the best of 128 attempts to fit the model are used to evaluate the model on the validation cohort and eventually the test cohort.

(TIF)

**S6 Fig. Comparison of applicable statistical classifiers on the repertoire classification problem.** Performance is measured using (i) log-likelihood, (ii) area under the curve (AUC) for true over false positive rates, (iii) balanced classification accuracy, and (iv) confidence cutoffs. The approach published by Emerson et al. achieves the best classification accuracy. We could not calculate the log-likelihood fit or apply confidence cutoffs because the source code to their method is unavailable. Instead, we use the results reported in their publication. The Max Snippet Model, as described in our earlier publications, has the worst performance [1–3]. The asterisk indicates the model reported in the main text. Using confidence cutoffs, we achieve a classification accuracy of 96%, capturing 18% of patients. We also performed a post-hoc hyperparameter optimization of our DKM model (last three rows) and found that we could achieve better results when the weights are arranged into 32 steps instead of 8 and for another DKM model where the weights are arranged into two sequences instead of one. Both these DKM models achieve better performances on the test cohort than the DKM model we report in the main text. However, we had already unblindfolded ourselves to the test cohort, so we did not report these result in the main text.

(TIF)

**S7 Fig. Plots of the confidence-correctness curves and sample retention curves used to visualize the confidence cutoffs.** (a) Using entropy as a measure of confidence, which is computed without labels, and cross entropy as a measure of correctness, which is computed with labels, we see a correlation between the confidence and correctness over each prediction (blue dots) of the statistical classifier on the antigen classification problem. Because the correlation exists, we can use the confidence to enrich for correctness, where the former is computed without knowing the labels. Samples captured to the right of the dashed line are classified with >95% accuracy (Fig 4), while samples to the left are considered indeterminant. No correlation exists for permuted data (red dots). (b) Sample retention curves show the classification accuracy as the confidence cutoff is increased, reducing the number of captured samples. Using the 95% cutoff computed using the validation cohort, samples from the test cohort are captured and the accuracy computed ("x"). (c) Like panel a, for the repertoire classification problem. (d) Like panel b, for the repertoire classification problem.

(TIF)

**S1 File. Additional description of the method under *dynamic kernel matching (supplemental)* and additional description of the results under *results (supplemental).***

(DOCX)

**S1 Data. Source code with instructions for downloading the data, building the datasets, and running the machine learning models.**
(ZIP)

## Acknowledgments

We are grateful that the antigen and repertoire classification datasets have been made available online by 10x Genomics and Adaptive Biotechnologies. Computing time on the UT Southwestern BioHPC computing cluster is made available through the Harold C. Simmons Comprehensive Cancer Center. This work is supported by the Department of Population and Data Sciences at UT Southwestern.

## Author Contributions

**Conceptualization:** Jared Ostmeyer, Scott Christley.

**Data curation:** Jared Ostmeyer.

**Formal analysis:** Jared Ostmeyer.

**Investigation:** Jared Ostmeyer.

**Methodology:** Jared Ostmeyer.

**Software:** Jared Ostmeyer.

**Validation:** Jared Ostmeyer.

**Visualization:** Jared Ostmeyer.

**Writing – original draft:** Jared Ostmeyer.

**Writing – review & editing:** Jared Ostmeyer, Lindsay Cowell.

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
