## [Decision Letter · Decision Letter 0]

9 Jun 2021

PONE-D-21-11812

Dynamic Kernel Matching for Non-conforming Data: A Case Study of T-cell Receptor Datasets

PLOS ONE

Dear Dr. Ostmeyer,

Thank you for submitting your manuscript to PLOS ONE. After careful consideration, we feel that it has merit but does not fully meet PLOS ONE’s publication criteria as it currently stands. Therefore, we invite you to submit a revised version of the manuscript that addresses the points raised during the review process.

We look forward to receiving your revised manuscript.

Kind regards,

Jiayin Wang, Ph.D.

Academic Editor

PLOS ONE

Journal Requirements:

Reviewers' comments:

Reviewer's Responses to Questions

**Comments to the Author**

1. Is the manuscript technically sound, and do the data support the conclusions?

Reviewer #1: Yes

Reviewer #2: Yes

2. Has the statistical analysis been performed appropriately and rigorously? 

Reviewer #1: Yes

Reviewer #2: Yes

3. Have the authors made all data underlying the findings in their manuscript fully available?

Reviewer #1: Yes

Reviewer #2: Yes

4. Is the manuscript presented in an intelligible fashion and written in standard English?

Reviewer #1: Yes

Reviewer #2: Yes

5. Review Comments to the Author

Reviewer #1: This manuscript provides an approach to find the potential pattern under the non-confirming data and the results demonstrate the proposed methods are useful in two non-conforming dataset. Overall, the article is well organized and its presentation is good. However, I think some minor issues still need to be improved:

1. Although the results show that a statistical classifier augmented with DKM can classify non-confirming data like sequences and sets of sequences and supplementatry figure 6 shows that DKM outperforms other traditional statistical classifiers, I think the author should pay more attention to the comparision results and my suggestions are as follows:

In order to extract features from TCR sequence which has not fixed sequence length, previous studies roughly have two types of practices in this field: align the sequences before calculating features, or using some special feature extraction methods (like k-mer features) to make the final feature dimensions the same. I think the author should describe the comparative experiment with more detail to reflect the effectiveness of the DKM method proposed in their article. For example, it would be better to provide the corresponding model parameters in detail.

2. Since the size of the training data may affect the final results. It would be better if the author can point out the size of the two datasets in detail in the Datasets section.

Reviewer #2: The authors propose a statistical classifier method DKM for processing non-conforming data, and use T-cell receptor (TCR) sequences labelled by disease antigen and patient cytomegalovirus (CMV) serostatus to conduct experiments and prove the effectiveness. This paper is full of work and has certain value in the research direction of immune database. However, several major questions are not clear to the reviewer:

Point 1: Page 5, line 81: “which uses the weights to place each feature into context based on its order in the sequence…” This only talked about the case where the data is a sequence. If the data is a sequence set, how to deal with it? Is the meaning of feature the same in sequence and sequence set? If they are not the same, please explain separately.

Point 2: Page 7, line 124; Page 7, line 139; Page 8, line 157: The meaning of Θ is not clear. The symbol described above represents amino acid. Here, it is said that Θ represents weight and symbol. Please make these expressions clearly, otherwise it will cause confusion of variables.

Point 3: Page 16, line 318-321: Please add the analysis of method comparison and analyze the reasons for the effectiveness of DKM method from the method itself.

6. PLOS authors have the option to publish the peer review history of their article (what does this mean?). If published, this will include your full peer review and any attached files.

Reviewer #1: No

Reviewer #2: No

---

## [Author Response · Author response to Decision Letter 0]

23 Jul 2021

RESPONSE TO REVIEWERS

Reviewer #1: This manuscript provides an approach to find the potential pattern under the non-confirming data and the results demonstrate the proposed methods are useful in two non-conforming dataset. Overall, the article is well organized and its presentation is good. However, I think some minor issues still need to be improved:

Reviewer #1: 1. Although the results show that a statistical classifier augmented with DKM can classify non-confirming data like sequences and sets of sequences and supplementatry figure 6 shows that DKM outperforms other traditional statistical classifiers, I think the author should pay more attention to the comparision results and my suggestions are as follows:

In order to extract features from TCR sequence which has not fixed sequence length, previous studies roughly have two types of practices in this field: align the sequences before calculating features, or using some special feature extraction methods (like k-mer features) to make the final feature dimensions the same. I think the author should describe the comparative experiment with more detail to reflect the effectiveness of the DKM method proposed in their article. For example, it would be better to provide the corresponding model parameters in detail.

Response: I added paragraphs to the antigen and repertoire classification results describing the comparisons in this study.

For the antigen classification problem, the discussion is focused on the two methods that achieve the best results (after DKM). The first method uses TCRDist as part of a nearest-neighbor classifier. Because TCRDist and DKM both use a sequence alignment algorithm, this paragraph highlights that TCRDist uses the BLOSUM62 matrix whereas DKM determines its similarity scores from the model weights. Contrasting the similarity scores helps the reader appreciate the alignment process with DKM.

The second method uses handcrafted features from a study by De Neuter et. al. There are many handcrafted features from De Neuter, and I am only able to provide a summary. Finally, I left in text pointing the reader to supplementary materials for additional comparisons.

In supplementary figure 6, I include the number of parameters from each model as you requested. Let me know if there is a method from a specific paper you would like me to include in my comparison and I will revise the manuscript.

Reviewer #1: 2. Since the size of the training data may affect the final results. It would be better if the author can point out the size of the two datasets in detail in the Datasets section.

Response: I added a description of the number of samples for each dataset.

For the antigen classification dataset, I list how many TCRs are associated with each pMHC in the dataset. The numbers below also appear in the manuscript under Datasets:

• GILGFVFTL:A0201 - 3982 TCRs

• KLGGALQAK:A0301 - 27561 TCRs

• RLRAEAQVK:A0301 - 612 TCRs

• IVTDFSVIK:A1101 - 2563 TCRs

• AVFDRKSDAK:A1101 - 3505 TCRs

• RAKFKQLL:B0801 - 5598 TCRs

As mentioned in the manuscript, we discard all pMHCs that are associated with fewer than 500 TCRs. I also added a sentence that we use a 60/20/20 split to build the training, validation, and test cohorts.

For the repertoire classification dataset, I list how many patients are associated with each label. The numbers below also appear in the manuscript under Datasets:

• CMV positive - 309

• CMV negative - 365

I also added a description of how the training, validation, and test cohorts were constructed.

Finally, I would like to highlight the number of features in the repertoire classification problem is large! For each sample there are over 100,000 TCRs, an average of 13 amino acid residues from the CDR3 of each TCR, and 5 Atchley factor features per amino acid residue. Thus, each sample has over 100,000 × 13 × 5 = 6.5 × 10^6 Atchley factor features. Rather than build a model with 6.5 million weights (one for each feature), we use DKM to match 40 weights to the 6.5 million Atchley factor features. It is amazing to me that DKM worked at all!

Thank you for reviewing my manuscript!

Jared

Reviewer #2: The authors propose a statistical classifier method DKM for processing non-conforming data, and use T-cell receptor (TCR) sequences labelled by disease antigen and patient cytomegalovirus (CMV) serostatus to conduct experiments and prove the effectiveness. This paper is full of work and has certain value in the research direction of immune database. However, several major questions are not clear to the reviewer:

Reviewer #2: Point 1: Page 5, line 81: “which uses the weights to place each feature into context based on its order in the sequence…” This only talked about the case where the data is a sequence. If the data is a sequence set, how to deal with it? Is the meaning of feature the same in sequence and sequence set? If they are not the same, please explain separately.

Response: I rewrote that text. I do not know if it is an improvement. If it is still too confusing, I will be happy to revise again.

To summarize,

• When dealing with sequences, each symbol is processed based on (i) the content of that symbol and (ii) the order of that symbol in the sequence.

• When dealing with sets, each symbol is processed based on (i) the content of that symbol and (ii) the presence (but not the order) of other symbols in the set.

• Now for sequence-sets, each symbol is processed based on (i) the content of that symbol, (ii) the order of that symbol in its sequence, and (iii) the presence (but not the order) of other sequences in the sequence-set.

Reviewer #2: Point 2: Page 7, line 124; Page 7, line 139; Page 8, line 157: The meaning of Θ is not clear. The symbol described above represents amino acid. Here, it is said that Θ represents weight and symbol. Please make these expressions clearly, otherwise it will cause confusion of variables.

Response: I revised the manuscript to remove any mention of the symbol capital Θ. I also cleaned up the text to avoid confusing language referring to Θ as representing weights and symbols. These changes required me to revise many sentences appearing later in the manuscript, but I think this is much clearer. Thanks!

Reviewer #2: Point 3: Page 16, line 318-321: Please add the analysis of method comparison and analyze the reasons for the effectiveness of DKM method from the method itself.

Response: I added a new paragraph describing the comparison of other methods. All comparisons are based on classification accuracy of the balanced datasets, which is not explicitly mentioned in the manuscript. To make space for this new paragraph, I moved the discussion about the permutation analysis to supplementary materials.

I left discussions about comparing DKM to slightly modified versions of the DKM model in supplementary. A previous reviewer wanting to see better results suggested modifications to the DKM model to improve its performance. However, I had already unblinded us to the test cohorts, so I did not feel comfortable reporting these results in the main manuscript. The comparison of DKM to itself is reported in supplementary where I make it clear that I had already unblinded us to the test cohort for these other versions of DKM.

Thank you for reviewing my manuscript!

Jared

---

## [Decision Letter · Decision Letter 1]

31 Aug 2021

PONE-D-21-11812R1

Dynamic Kernel Matching for Non-conforming Data: A Case Study of T Cell Receptor Datasets

PLOS ONE

Dear Dr. Ostmeyer,

Thank you for submitting your manuscript to PLOS ONE. After careful consideration, we feel that it has merit but does not fully meet PLOS ONE’s publication criteria as it currently stands. Therefore, we invite you to submit a revised version of the manuscript that addresses the points raised during the review process.

We look forward to receiving your revised manuscript.

Kind regards,

Jiayin Wang, Ph.D.

Academic Editor

PLOS ONE

Additional Editor Comments:

Please carefully consider the comments from reviewers

Reviewers' comments:

Reviewer's Responses to Questions

**Comments to the Author**

1. If the authors have adequately addressed your comments raised in a previous round of review and you feel that this manuscript is now acceptable for publication, you may indicate that here to bypass the “Comments to the Author” section, enter your conflict of interest statement in the “Confidential to Editor” section, and submit your "Accept" recommendation.

Reviewer #3: (No Response)

2. Is the manuscript technically sound, and do the data support the conclusions?

Reviewer #3: Partly

3. Has the statistical analysis been performed appropriately and rigorously? 

Reviewer #3: Yes

4. Have the authors made all data underlying the findings in their manuscript fully available?

Reviewer #3: Yes

5. Is the manuscript presented in an intelligible fashion and written in standard English?

Reviewer #3: Yes

6. Review Comments to the Author

Reviewer #3: The author suggested that the TCR sequencing data was a non-conforming data and proposed a dynamic kernel matching method to handle non-conforming data of two datasets. The author considered that the DKM model could be applied to the antigen classification and repertoire classification problems, respecitvely. This paper had certain amount of work, while lack of innovation. The author experimented the existing DKM model on the public database, and obtained the prediction results. However, there were still some problems in this study.

1. In the line 258, the author told us “there were 44 pMHCs in the database, while only the six pMHCs could interacted with the receptors”, how about the other 38 pMHCs? Maybe some of the them could be used for the antigen classification, but the accuracy of the algorithm in not high enough to predict them, or the author should use the other methods, such as the biological experimental, to verified the other 38 pMHCs had not been used for antigen classification.

2. In line 371, “the classification accuracy of DKM is significantly worse than Emerson’s method on the repertoire classification problem”, in author’s opinion, they thought the advantage of DKM was that it could predict the antigen classification and repertoire classification simultaneously, but the prediction results of repertoire would definitely affect the subsequent results. Furthermore, if the time complexity was not high enough, other researchers could run the two different models to obgtain the classification results of antigen and repertoire, to replace the DKM. Thus, to some extent, the author should modified the DKM model, such as some parameters or others, to improve the accuracy in predicting antigen and repertoire classification.

3. When compared the DKM model with other methods, in addition to the prediction accuracy, the author should use more indicators to explained them comprehensively.

7. PLOS authors have the option to publish the peer review history of their article (what does this mean?). If published, this will include your full peer review and any attached files.

Reviewer #3: No

---

## [Author Response · Author response to Decision Letter 1]

26 Oct 2021

Reviewer #3: In the line 258, the author told us “there were 44 pMHCs in the database, while only the six pMHCs could interacted with the receptors”, how about the other 38 pMHCs? Maybe some of the them could be used for the antigen classification, but the accuracy of the algorithm in not high enough to predict them, or the author should use the other methods, such as the biological experimental, to verified the other 38 pMHCs had not been used for antigen classification.

Response: For the antigen classification problem, all the models have exactly six outputs, one for each pMHC. For this reason, our comparisons of DKM to other models do not generalize beyond these six pMHCs. Given this constraint, we cannot utilize the remaining 38 pMHCs to further compare DKM to other approaches.

To train a model to categorize each TCR by pMHC, examples of TCRs interacting with each pMHC are required. Of the 44 pMHCs in the dataset, only the six pMHCs interacting with at least 500 unique receptors are used because only these six pMHCs provide enough examples to train a categorical model. With a minimum of 500 TCRs per pMHC, we could split the TCRs into a training, validation, and test cohort.

I initially considered a generative model (as opposed to a categorical model) that could predict the interactions of any TCR with any pMHC. However, my initial analyses determined any generative model would require far more than the 44 pMHC examples available in the antigen classification dataset. As more pMHCs become available in the future, we will reevaluate if we can construct a generative model for all pMHCs.

Reviewer #3: In line 371, “the classification accuracy of DKM is significantly worse than Emerson’s method on the repertoire classification problem”, in author’s opinion, they thought the advantage of DKM was that it could predict the antigen classification and repertoire classification simultaneously, but the prediction results of repertoire would definitely affect the subsequent results. Furthermore, if the time complexity was not high enough, other researchers could run the two different models to obgtain the classification results of antigen and repertoire, to replace the DKM. Thus, to some extent, the author should modified the DKM model, such as some parameters or others, to improve the accuracy in predicting antigen and repertoire classification.

Response: In response to your suggestion, I added another hyperparameter for the DKM model on the repertoire classification task. For this hyperparameter, I experimented with trimming CDR3 sequences because previous studies have shown the beginning and end of the CDR3 sequence are not involved in antigen recognition [1,2]. Unfortunately, trimming CDR3 sequences did not improve performance (supplementary figure 10).

For a list of various DKM models considered in this study, refer to supplementary figure 10. Some of the DKM variations performed better than the original model as described in the manuscript, but none of the models outperformed the original results reported by Emerson et al. I left the discussion of DKM variants in supplementary material because I had already unblinded us to the test cohort, so I did not feel comfortable reporting these results in the main manuscript.

I am unable to perform an exhaustive hyperparameter search because my resources are allocated by funded projects.

Reviewer #3: When compared the DKM model with other methods, in addition to the prediction accuracy, the author should use more indicators to explained them comprehensively.

Response: In response to your suggestion, we added area under the curve (AUC) for plots of the true versus false positive rates of each model. The AUC indicates how much the true positive rate is better than the false positive rate. An AUC of 1 indicates perfect predictive performance whereas an AUC of 0.5 indicates no predictive performance. The AUC is an excellent metric for measuring predictive performance on imbalanced datasets like ours, which is why I selected this metric. Because the AUC compares the true to false positive rates, it is designed for binary classifiers. To apply the AUC to a multinomial model with 6 outputs, we treated each output as a binary classifier using a one-vs-rest criteria, like others have done, by averaging the AUC for all 6 outputs [3].

On the antigen classification problem, our DKM model achieves an AUC of 0.91 (see supplementary figure 6), outperforming all other approaches. Our DKM model achieves the highest AUC of all the models for the antigen classification task. Indeed, our DKM model achieves the best performance across the three standard metrics we considered (KL-divergence, balanced classification accuracy, and now AUC).

On the repertoire classification problem, our DKM model achieves an AUC of 0.75 (see supplemental figures 9, 10). While an AUC of 0.75 is okay, it is still below the AUC of 0.94 reported by Emerson et al.

In addition to the AUC, we report the KL-divergence (or just the log-likelihood when appropriate) and balanced classification accuracy, providing three standard metrics for interpreting performance. If there is a specific metric you would like us to consider, please suggest it to us in your response. Bear in mind that many metrics are limited to a single type of statistical classifier and do not generally apply to both binary and multinomial predictors.

Thank you!

Jared

References:

1. Ostmeyer, Jared, et al. "Biophysicochemical motifs in T-cell receptor sequences distinguish repertoires from tumor-infiltrating lymphocyte and adjacent healthy tissue." Cancer research 79.7 (2019): 1671-1680.

2. Glanville, Jacob, et al. "Identifying specificity groups in the T cell receptor repertoire." Nature 547.7661 (2017): 94-98.

3. De Neuter, Nicolas, et al. "On the feasibility of mining CD8+ T cell receptor patterns underlying immunogenic peptide recognition." Immunogenetics 70.3 (2018): 159-168.

---

## [Decision Letter · Decision Letter 2]

1 Dec 2021

PONE-D-21-11812R2Dynamic Kernel Matching for Non-conforming Data: A Case Study of T Cell Receptor DatasetsPLOS ONE

Dear Dr. Ostmeyer,

Thank you for submitting your manuscript to PLOS ONE. After careful consideration, we feel that it has merit but does not fully meet PLOS ONE’s publication criteria as it currently stands. Therefore, we invite you to submit a revised version of the manuscript that addresses the points raised during the review process.

We look forward to receiving your revised manuscript.

Kind regards,

Jiayin Wang, Ph.D.

Academic Editor

PLOS ONE

Journal Requirements:

Additional Editor Comments (if provided):

Hope the rest issues could be solved easily.

Reviewers' comments:

Reviewer's Responses to Questions

**Comments to the Author**

1. If the authors have adequately addressed your comments raised in a previous round of review and you feel that this manuscript is now acceptable for publication, you may indicate that here to bypass the “Comments to the Author” section, enter your conflict of interest statement in the “Confidential to Editor” section, and submit your "Accept" recommendation.

Reviewer #3: All comments have been addressed

Reviewer #4: (No Response)

2. Is the manuscript technically sound, and do the data support the conclusions?

Reviewer #3: Yes

Reviewer #4: Yes

3. Has the statistical analysis been performed appropriately and rigorously? 

Reviewer #3: Yes

Reviewer #4: Yes

4. Have the authors made all data underlying the findings in their manuscript fully available?

Reviewer #3: Yes

Reviewer #4: Yes

5. Is the manuscript presented in an intelligible fashion and written in standard English?

Reviewer #3: Yes

Reviewer #4: Yes

6. Review Comments to the Author

Reviewer #3: (No Response)

Reviewer #4: This is an innovative study, described an approach (named dynamic kernel matching (DKM)) for modifying established statistical classifiers to handle non-conforming data. TCR sequences are one example of a datatype with non-conforming features. Two datasets of TCR sequences labelled by disease antigen and patient cytomegalovirus (CMV) serostatus were applied in the study. The statistical classifier performed well in the antigen classification dataset (the accuracy is 67.4%-70.5%). As for repertoire classification dataset, the accuracy is 67.6%, lower than a previously reported result by Emerson and colleagues. The author responded well to the reviewer#3’s queries.

Following are considerations for further improvement of the manuscript at the discretion of the authors and editors:

A) In the Methods & Materials, the author described the weights in DKM model. were the frequency of each TCR clone included in the weight calculation?

B) How does the author homogenize and normalize the TCR sequences dataset of patient cytomegalovirus (CMV) serostatus? Especially, the TCR clones with low frequency could be sequencing errors.

C) In the line 403, the author told us “For missing features, like patient age, we use a value of zero”, Does this affect the accuracy of the classifier? All patients’ age was number greater than 0, and age is an important factor affecting immunity.

D) The ROC curves of the test cohort could be put into the result of article.

7. PLOS authors have the option to publish the peer review history of their article (what does this mean?). If published, this will include your full peer review and any attached files.

Reviewer #3: No

Reviewer #4: **Yes: **Changxi Wang

---

## [Author Response · Author response to Decision Letter 2]

14 Jan 2022

Reviewer #4: This is an innovative study, described an approach (named dynamic kernel matching (DKM)) for modifying established statistical classifiers to handle non-conforming data. TCR sequences are one example of a datatype with non-conforming features. Two datasets of TCR sequences labelled by disease antigen and patient cytomegalovirus (CMV) serostatus were applied in the study. The statistical classifier performed well in the antigen classification dataset (the accuracy is 67.4%-70.5%). As for repertoire classification dataset, the accuracy is 67.6%, lower than a previously reported result by Emerson and colleagues. The author responded well to the reviewer#3’s queries.

Following are considerations for further improvement of the manuscript at the discretion of the authors and editors:

Reviewer #4: In the Methods & Materials, the author described the weights in DKM model. were the frequency of each TCR clone included in the weight calculation?

Response: The clonal frequencies are used indirectly to calculate the weights for the antigen and repertoire classification models. The clonal frequency is introduced into the calculations of the weights during the fitting procedure whenever we calculate a mean or standard deviation. We believe it is most appropriate to weight the terms in the mean and standard deviations by the clonal frequency. Means and standard deviations are calculated, for example, whenever we scale terms in the model using batch normalization (see Supplementary Text, Dynamic Kernel Matching, Scaling and Supplemental Figures 3 and 7).

For the antigen classification problem, where each clone represents an individual sample, we use the clonal frequency to weight each sample in the loss function (objective function). This is an additional source by which the clonal frequency indirectly influences the calculation of the weights of the DKM model specific to the antigen classification model.

Reviewer #4: How does the author homogenize and normalize the TCR sequences dataset of patient cytomegalovirus (CMV) serostatus? Especially, the TCR clones with low frequency could be sequencing errors.

Response: The numeric representations of the TCR sequences are passed to the model without being normalized. However, after multiplying each feature with a weight, we normalize the subsequent terms by batch normalization (see Supplementary Text, Dynamic Kernel Matching, Scaling and Supplemental Figures 3 and 7). This ensures the expected magnitude of each term is the same at the start of the gradient optimization procedure. Of note is that we wrote our own batch normalization code rather than use TensorFlow code because the existing code did not support weighted calculations of the mean and variance. To simplify our code for batch normalization, we only apply batch normalization at the start of but not during gradient optimization.

While we expect some TCR sequences to contain a sequencing error, the vast majority will not. We ignore sequencing error when predicting the CMV serostatus because sequencing error effects only a small number of the features and not the label, which is based on serostatus.

Reviewer #4: In the line 403, the author told us “For missing features, like patient age, we use a value of zero”, Does this affect the accuracy of the classifier? All patients’ age was number greater than 0, and age is an important factor affecting immunity.

Response: I failed to communicate what we are doing at line 403, so I rewrote that text to clarify what we did.

We were not attempting to fill in missing features for the repertoire classification problem (the patient age is known for every sample). Rather, we were attempting to determine if a DKM model fitted to the repertoire classification problem is forced to learn how to identify TCR sequences with CMV specificity. To test this, we fit the DKM model to the repertoire classification problem and applied the weights to score individual TCR sequences with known specificity. However, some of the features in the repertoire classification problem, like patient age, are not features of the TCR sequences with known specificity. Therefore, we assigned all these features a value of 0, which is fine because we are trying to score individual TCRs rather than predict a label for an entire repertoire.

Reviewer #4: The ROC curves of the test cohort could be put into the result of article.

Response: For the repertoire classification problem, I updated figure 3 to contain a ROC curve. For the antigen classification problem, I updated figure 2 with a confusion matrix because we treated the antigen classification problem as a multinomial classification problem. Alternatively, we could have treated the multinomial classifier as many separate binary classifiers and generate ROC curves for each binary classifier. Ultimately, we believed the reader would have a harder time interpreting a series of ROC curves generated from a single multinomial model, which is why we chose to go with a confusion matrix instead.

Thank you, Dr. Wang.

---

## [Decision Letter · Decision Letter 3]

1 Mar 2022

Dynamic Kernel Matching for Non-conforming Data: A Case Study of T Cell Receptor Datasets

PONE-D-21-11812R3

Dear Dr. Ostmeyer,

We’re pleased to inform you that your manuscript has been judged scientifically suitable for publication and will be formally accepted for publication once it meets all outstanding technical requirements.

Kind regards,

Jiayin Wang, Ph.D.

Academic Editor

PLOS ONE

Additional Editor Comments (optional):

Dear authors, please consider all of the suggestions in the final manuscript.

Reviewers' comments:

Reviewer's Responses to Questions

**Comments to the Author**

1. If the authors have adequately addressed your comments raised in a previous round of review and you feel that this manuscript is now acceptable for publication, you may indicate that here to bypass the “Comments to the Author” section, enter your conflict of interest statement in the “Confidential to Editor” section, and submit your "Accept" recommendation.

Reviewer #5: All comments have been addressed

Reviewer #6: (No Response)

2. Is the manuscript technically sound, and do the data support the conclusions?

Reviewer #5: Yes

Reviewer #6: Yes

3. Has the statistical analysis been performed appropriately and rigorously? 

Reviewer #5: Yes

Reviewer #6: Yes

4. Have the authors made all data underlying the findings in their manuscript fully available?

Reviewer #5: Yes

Reviewer #6: Yes

5. Is the manuscript presented in an intelligible fashion and written in standard English?

Reviewer #5: Yes

Reviewer #6: Yes

6. Review Comments to the Author

Reviewer #5: In my opinion, the authors have addressed all comments of the reviewers. One tiny comment: the authors might want to cite https://doi.org/10.1016/j.coisb.2020.10.010 as it the most review on all things concerning AIRR classification.

Reviewer #6: The authors describe an approach, dynamic kernel matching or DKM, for handling variable-length features. In the case of ordered sequences of features, the key idea is to use a sequence-alignment style algorithm to match input features to weights. For sets of features, an assignment-problem algorithm is applied to match features to weights. Overall, the manuscript is well written and the associated software on github seems well documented and usable. I have only minor suggestions:

* the main text does not do a great job of describing the actual regression models used, for example the fact that many other features such as V/J gene usage and CDR3 length are incorporated: I would vote to move Figs S3 and S7 into the main text, or at least mention in words the features going into the model. I also would really love to know how well the antigen-classification model works if the DKM modules are ablated, since V/J genes and length convey a fair amount of information on their own.

* Since the number of weight "sequences" and the length of the weight sequences are key aspects of the DKM model, it would be nice to see this documented in the paper. From the Supplement, it looks like a range of values for both were tried? 1/2 sequences, 4/8/32 for the lengths. These numbers are important in thinking about what the alignment procedure is doing: can we think of it as aligning CDR3s to an "optimal" CDR3 sequence, or a concatenation of optimal subsequences? Granted that the weights are not necessarily mappable to amino acid symbols, but that seems like a possible intuition vis-a-vis sequence alignment. Related, for the repertoire classification, it looks like the set assignment matching is actually just a "max", ie there's only one weight-sequence. If we envision that CMV-specific TCRs fall into multiple different specificity groups, is this "max" just looking for a single, most-predictive specificity group? Could performance be improved by including more weight sequences for matching?

A few little typos:

"with pMHC (≥5Å), with" should this be <= 5?

"complimentary determining regions 3 (CDR3s)"  "complementarity-determining region"

"sets of made sequences"  sets made of sequences

7. PLOS authors have the option to publish the peer review history of their article (what does this mean?). If published, this will include your full peer review and any attached files.

Reviewer #5: No

Reviewer #6: No

---

## [Editor Report · Acceptance letter]

25 Mar 2022

PONE-D-21-11812R3 

Dynamic Kernel Matching for Non-conforming Data:
A Case Study of T Cell Receptor Datasets 

Dear Dr. Ostmeyer:

I'm pleased to inform you that your manuscript has been deemed suitable for publication in PLOS ONE. Congratulations! Your manuscript is now with our production department. 

Kind regards, 

on behalf of

Dr. Jiayin Wang 

Academic Editor

PLOS ONE